# Hyperspectral Imaging Combined with Artificial Intelligence in the Early Detection of Esophageal Cancer

**DOI:** 10.3390/cancers13184593

**Published:** 2021-09-13

**Authors:** Cho-Lun Tsai, Arvind Mukundan, Chen-Shuan Chung, Yi-Hsun Chen, Yao-Kuang Wang, Tsung-Hsien Chen, Yu-Sheng Tseng, Chien-Wei Huang, I-Chen Wu, Hsiang-Chen Wang

**Affiliations:** 1Division of Gastroenterology and Hepatology, Department of Internal Medicine, Ditmanson Medical Foundation Chiayi Christian Hospital, Chia Yi City 60002, Taiwan; momotsai5@gmail.com (C.-L.T.); jayshung1985@gmail.com (Y.-H.C.); cych13794@gmail.com (T.-H.C.); 2Department of Mechanical Engineering, Advanced Institute of Manufacturing with High tech Innovations (AIM-HI) and Center for Innovative Research on Aging Society (CIRAS), National Chung Cheng University, 168, University Rd., Min Hsiung, Chia Yi County 62102, Taiwan; arvindmukund96@gmail.com (A.M.); lonesome310160@hotmail.com (Y.-S.T.); 3Department of Internal Medicine, Far Eastern Memorial Hospital, No.21, Sec. 2, Nanya S. Rd., Banciao Dist., New Taipei City 22060, Taiwan; chungchenshuan3@yahoo.com.tw; 4Division of Gastroenterology, Department of Internal Medicine, Kaohsiung Medical University Hospital, Kaohsiung Medical University, No.100, Tzyou 1st Rd., Sanmin Dist., Kaohsiung City 80756, Taiwan; fedwang@gmail.com (Y.-K.W.); minicawu@gmail.com (I.-C.W.); 5Department of Medicine, Faculty of Medicine, College of Medicine, Kaohsiung Medical University, No.100, Tzyou 1st Rd., Sanmin Dist., Kaohsiung City 80756, Taiwan; 6Graduate Institute of Clinical Medicine, College of Medicine, Kaohsiung Medical University, No.100, Tzyou 1st Rd., Sanmin Dist., Kaohsiung City 80756, Taiwan; 7Department of Gastroenterology, Kaohsiung Armed Forces General Hospital, 2, Zhongzheng 1st. Rd., Lingya District, Kaohsiung City 80284, Taiwan; 8Department of Nursing, Tajen University, 20, Weixin Rd., Yanpu Township, Pingtung City 90741, Taiwan

**Keywords:** hyperspectral imaging, single-shot multibox detector, white-light endoscopy, narrow-band endoscopy and dysplasia

## Abstract

**Simple Summary:**

Detection of early esophageal cancer is important to improve patient survival, however, early diagnosis of the cancer cells is difficult, even for experienced endoscopists. This article provides a new method by using hyperspectral imaging and a deep learning diagnosis model to classify and diagnose esophageal cancer using a single-shot multibox detector. The accuracy of the results when using an RGB image in WLI was 83% and while using the spectrum data the accuracy was increased to 88%. There was an increase of 5% in WLI. The accuracy of the results when using an RGB image in NBI was 86% and while using the spectrum data the accuracy was increased to 91%. There was an increase of 5% in NBI. This study proves that the accuracy of prediction when using the spectrum data has been significantly improved and the diagnosis of narrow-band endoscopy data is more sensitive than that of white-light endoscopy.

**Abstract:**

This study uses hyperspectral imaging (HSI) and a deep learning diagnosis model that can identify the stage of esophageal cancer and mark the locations. This model simulates the spectrum data from the image using an algorithm developed in this study which is combined with deep learning for the classification and diagnosis of esophageal cancer using a single-shot multibox detector (SSD)-based identification system. Some 155 white-light endoscopic images and 153 narrow-band endoscopic images of esophageal cancer were used to evaluate the prediction model. The algorithm took 19 s to predict the results of 308 test images and the accuracy of the test results of the WLI and NBI esophageal cancer was 88 and 91%, respectively, when using the spectral data. Compared with RGB images, the accuracy of the WLI was 83% and the NBI was 86%. In this study, the accuracy of the WLI and NBI was increased by 5%, confirming that the prediction accuracy of the HSI detection method is significantly improved.

## 1. Introduction

Esophageal cancer can be regarded as the least researched and one of the deadliest cancers around the globe [1,2]. It is the eighth most common cancer and the sixth most common cause of cancer death [3]. At present, the endoscopists are unable to draw conclusions from the images of esophageal cancer. This also causes early symptoms to be easily overlooked. A precise assessment for individualized treatment depends on the accuracy of the initial diagnosis [4]. The images obtained by the means of general instruments will be affected by tissue secretions or instrument specifications, which may directly or indirectly lead to misjudgments in diagnosis. However, by using hyperspectral imaging technology combined with artificial intelligence deep learning methods to perform spectral data for esophageal cancer will offer a faster and more accurate diagnosis. The hyperspectral images have nanometer-level spectral intervals and hence the amount of spectrum information that can be detected is much larger than that of multispectral images [5,6,7,8,9,10,11]. The image spectrum conversion is the use of an imaging spectrometer to obtain an image that has a wide wavelength measurement range, the image is divided into a multispectral image and a hyperspectral image according to the spectral resolution. The two images are usually separated by a resolution of 10 nm. When the spectral resolution of the image is less than or equal to 10 nm, it is a hyperspectral image; when the spectral resolution of the image is greater than 10 nm, it is called a multispectral image. In addition, the spectral characteristics are unique to each substance, so it can be used to identify any object, which can be very accurately identified from the spectral characteristics or spectral response of the material or object in the obtained hyperspectral image [12,13]. In recent years, many scholars have conducted deep learning-related research on esophageal cancer which has proved to be a potential method for the early detection of esophageal cancer [14,15,16,17,18,19,20,21]. Shahidi et al. conducted research on white-light endoscopic images and narrow-band endoscopic images of gastrointestinal cancers to accurately diagnose and treat the esophagus by artificial intelligence [22]. Yoshitaka et al. conducted research on the application of convolutional neural networks in artificial intelligence to determine the invasion depth of esophageal cancer under white-light endoscopy. They also proved that the AI diagnostic system is much more effective for the diagnostic accuracy of esophageal squamous cell carcinoma (ESCC) as the depth of invasion is higher than that of endoscopists [23]. Hiromu et al. also studied the comparison between artificial intelligence and endoscopy professional physicians in real-time assisted diagnosis of ESCC and proved that the AI method had a higher detection sensitivity and accuracy of noncancerous tissue [24]. Wang et al. developed a single-shot multibox detector using a convolutional neural network for diagnosing esophageal cancer by using endoscopic images.

The main objective of this research is to use the hyperspectral conversion of the esophageal cancer image to obtain the spectrum information and then use the SSD model to perform deep learning training on the esophageal cancer spectrum information, thereby establishing the detection model for diagnosis of the stage and the location of cancerous lesions.

## 2. Materials and Methods

The number of endoscopic images of esophageal cancer used in this study was 1232. Out of these images, the number of WLI and NBI esophageal cancers was 620 and 612, respectively. Among the 620 white-light endoscopic images, the number of images co-relating with four levels of severity were 100, 112, 196 and 212 normal, low-grade dysplasia, high-grade dysplasia, invasive cancer, respectively. Among the 612 narrow-band esophageal cancer endoscopy images, the number of images co-relating with four stages in esophageal cancer were 100, 108, 180 and 224 normal, low-grade dysplasia, high-grade dysplasia, invasive cancer, respectively. In this study, the system was trained by 1232 images of esophageal cancer and its precursor lesions (620 WLI, 612 NBI), and then 308 images (155 WLI, 153 NBI) were evaluated with the trained system (accuracy of AI-HIS). The hyperspectral conversion technology (Smart Spectrum Cloud, Hitspectra Intelligent Technology Co., Ltd., Kaohsiung City, Taiwan) used in this study to convert the two types of endoscopic esophageal cancer images into 401 bands (380–780 nm) is shown in Figure 1.

The Deep Convolutional Neural Network (CNN) model of the Single Shot MultiBox Detector (SSD, Hitspectra Intelligent Technology Co., Ltd., Kaohsiung City, Taiwan) used in this study is a detection architecture based on the VGG-16-Atrous network [25]. SSD mainly predicts from multiple feature maps [16]. Vgg-16 is a deep learning architecture with 16 hidden layers composed of 13 convolutional layers and 3 fully connected layers [26,27]. The key feature of this model is that it is at least one order of magnitude more efficient than the existing methods and has a default box with different positions, scales, aspect ratios, and a multiscale convolutional bounding box output using multiple feature maps [28]. The principle of SSD for target matching will first calculate the intersection over union (IOU) value between the default frame and the real target, and then will select the default frame with the largest IOU value to match the real frame. The default frame for successful matching is the positive sample. Negative samples that are successfully matched will be evaluated again. If the IOU value of the default box is not successfully matched with the real box having greater than a certain threshold (generally set to 0.5), it will be reclassified as a positive sample. For multiple feature maps, SSD uses a set of preset default bounding boxes to associate with each feature mapping unit, and the default bounding boxes are tiled on the feature map in a convolutional manner. Therefore, the position of each bounding box relative to its corresponding unit is fixed. The offset relative to the default bounding box shape in each feature map is predicted and the category of each object existing in each bounding box score from the training model loss can be calculated from Equation (1) [29].
(1)$$L\left(x,c,l,g\right)=\frac{1}{N}\left(L_{c}onf\left(x,c\right)+\alpha L_{l}oc\left(x,l,g\right)\right)$$

*N* is the number of positive samples of the matching default bounding box. If *N* = 0, the loss value is defined as 0; *c* is the predicted value of confidence loss; *l* is the predicted position of the bounding box corresponding to the default box; *g* is the position of ground truth parameter. In this study, the spectrum information of esophageal cancer and the label information that predicts the stage and coordinates of the disease are used in the SSD detection model framework. The data batch size is 32, the global learning rate is 0.0005, and the total number of training steps is 80,000. The white-light endoscopic imaging of esophageal cancer data and the narrow-band endoscopic imaging of esophageal cancer data are separately trained. The complete flow chart of this study is shown in Figure 2.

The spectrum information obtained by image spectrum conversion has 401 characteristic dimensions in the visible light band (380–780 nm). Figure 3a,c shows the spectrum of esophageal cancer in four stages: normal, low-grade dysplasia, high-grade dysplasia, invasive cancer of white-light endoscopic images and narrow-band endoscopic images, respectively. The high-dimensional spectrum information of the white-light and narrow-band endoscopes is reduced to three-dimensions using principal component analysis (PCA) [30]. The two kinds of endoscope spectrum information after dimension can show good variability between the severity of esophageal cancer as shown in Figure 3b,d.

The color image obtained by the white-light and narrow-band endoscope needed to be subjected to a series of preprocessing. In order to increase the training data, the preprocessed images were augmented, and the label information of the data was created based on this, which contained the name and coordinate location of the disease stage. On the other hand, using 24 color blocks as the reference object, a spectrometer was used to establish an image spectrum conversion module for the endoscope. By using the multivariate regression analysis, the conversion matrix that converted the endoscopic images into the spectrum of the visible light band could be obtained. With the conversion matrix, the images could be converted into the spectrum of the visible light band to obtain the spectrum information of the esophageal cancer. Next, the spectrum information of esophageal cancer was reduced by principal component analysis. Finally, by using SSD we trained a diagnostic model to predict the number and location of esophageal cancers, based on frequency spectrum.

In this study, 155 white-light esophageal cancer spectrum data and 153 narrow-band esophageal cancer spectrum information were used to test the white-light prediction model and the narrow-band prediction model, respectively. The resulting prediction model can be presented at the same time as the corresponding original esophageal cancer image. This kind of visualization allowed us to observe and evaluate the prediction results with a very intuitive presentation effect. The results displayed a rectangular box surrounding the esophageal cancer lesion in the endoscopic image. The predicted rectangular box represented a corresponding color according to the assigned disease stage name; dysplasia corresponds to low-grade dysplasia, high-grade dysplasia corresponds to light blue-gray, invasive cancer corresponds to orange, and for normal there would not be any prediction box. The kappa statistic is frequently used to test interrater reliability [31]. The importance of reliability lies in the fact that it represents the extent to which the data collected in the study are correct representations of the variables measured. Like most correlation statistics, the kappa can range from −1 to +1. The kappa is one of the most commonly used statistics to test interrater reliability [32].

## 3. Results

Figure 4 shows the image results of esophageal cancer prediction presented on a white-light endoscopic imaging while Figure 5 shows the image results of esophageal cancer prediction presented on a narrow-band endoscope.

Table 1 compares the test results of the esophageal cancer spectrum data in this study with the test results using RGB esophageal cancer images. The table sorts out the spectrum data obtained by white-light endoscopes and narrow-band endoscopes. The test data results of RGB images are shown under four severity levels. The experimental data was statistically based on the confusion matrix and the results are shown in Table 2. In the white-light endoscopy spectrum data, out of the 155 images, 25, 28, 49 and 53 images represent the normal, low-grade dysplasia, high-grade dysplasia, invasive cancer case respectively. Among the prediction results of the four cancer stages, 23 in normal, 21 in low-grade dysplasia, 45 in high-grade dysplasia, and 48 in invasive cancer were predicted correctly. Therefore, the accuracy of the white-light endoscopy data test result was 88%. On the other hand, in the narrow-band endoscopy spectrum data, out of the images, 25, 27, 48 and 56 images represent the normal, low-grade dysplasia, high-grade dysplasia, invasive cancer case respectively. Among the prediction results of the four cancer stages, 24 in normal, 24 in low-grade dysplasia, 41 in high-grade dysplasia, and 51 in invasive cancer were predicted correctly. Therefore, the accuracy of the white-light endoscopy data test result was 91%. As seen from Table 2, the sensitivity rate of WLI of normal stage in RGB is 76% while for the same stage in the WLI spectrum the sensitivity was 92%. This proves that the efficiency in detecting the esophageal cancer in the normal stage has been improved by 16%. In the low-grade dysplasia stage, the sensitivity rate of WLI in RGB was 61% while for the same stage in the WLI spectrum the sensitivity was 75%. This drop in the sensitivity rate was due to fewer sample rates being available for this stage. However, in the high-grade dysplasia and invasive cancer stages the sensitivity rate in WLI RGB was 86% and 93%, respectively, while the WLI spectrum was 92 and 91%, respectively. The same pattern could be found in the NBI band also. As seen from Table 2, the sensitivity rate of NBI of normal stage in RGB was 68%, while for the same stage in the NBI spectrum the sensitivity was 96%. This proves that, similar to the WLI band, the efficiency of detecting the esophageal cancer in the early stage by using NBI spectrum was improved by 28%. In the low-grade dysplasia stage, the sensitivity rate of NBI in RGB was 83%, while for the same stage in the NBI spectrum the sensitivity was 89%. Unlike the WLI band, the sensitivity rate was increased by 6%. From this interference it is clear that for the first two stages of detecting esophageal cancer, the NBI band can be used, while for the latter two stages, the WLI band can be used. However, in the high-grade dysplasia and invasive cancer stages the sensitivity rate in NBI RGB was 88 and 98%, respectively while in the NBI spectrum, both were 91%. As seen in Table 2, the kappa WLI RBG and NBI RGB was 0.76 and 0.81 while the kappa for the WLI spectrum and NBI spectrum was 0.84. This shows that there was an increase of 0.8 and 0.3 in kappa for the WLI and NBI spectra.

## 4. Discussion

From this article, it is very clear that by using a hyperspectral imaging (HSI) and deep learning diagnosis model the accuracy in the early detection of esophageal cancer can be increased significantly. This study provides SSD, a fast single-shot multi-class object detector. The model is characterized by a multiscale convolutional bounding box output using multiple feature maps. The position, scale and aspect ratio of the predicted sample were greater by at least one order of magnitude when compared with the existing methods. This advantage enables the model to train the bounding box shape space more effectively. The algorithm took 19 s to predict the results of 308 test images. The accuracy of the results when using RGB image in WLI was 83% and while using the spectrum data the accuracy was increased to 88%. There was an increase of 5% in WLI. The accuracy of the results when using RGB image in NBI was 86% and while using the spectrum data the accuracy was increased to 91%. There was an increase of 5% in NBI. At present, there are not many studies on spectrum information of esophageal cancer. Even so, by increasing the amount of data for training spectrum information, the accuracy of model diagnosis can be improved significantly. This study was conducted at the Kaohsiung Medical University and a total of 45 patients took part in this study. Currently this method has not been used in any clinical diagnosis. However, in the future, if research on the band selection for spectrum information of esophageal cancer is proposed in the future, using this method to reduce the amount of spectrum information, combined with the detection and diagnosis methods of this study will be useful for the early diagnosis of esophageal cancer and will have better predictions.

## 5. Conclusions

This study proves that the accuracy of prediction when using spectrum data has been significantly improved and the diagnosis of narrow-band endoscopy data is more sensitive than that of white-light endoscopy.

## Figures and Tables

**Figure 1 cancers-13-04593-f001:**
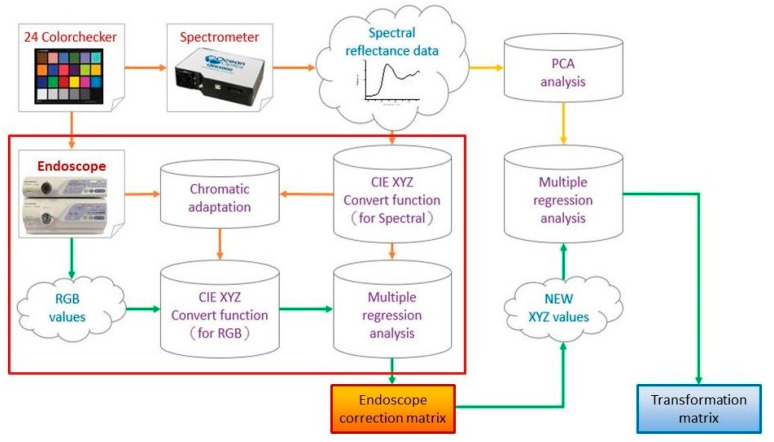
The flow chart of spectrum conversion construction, using standard 24 color blocks (X-Rite Classic, 24 Color Checkers) as the common target object for spectrum conversion of the endoscope and spectrometer, converting the endoscopic images into 401 bands of visible light spectrum information.

**Figure 2 cancers-13-04593-f002:**
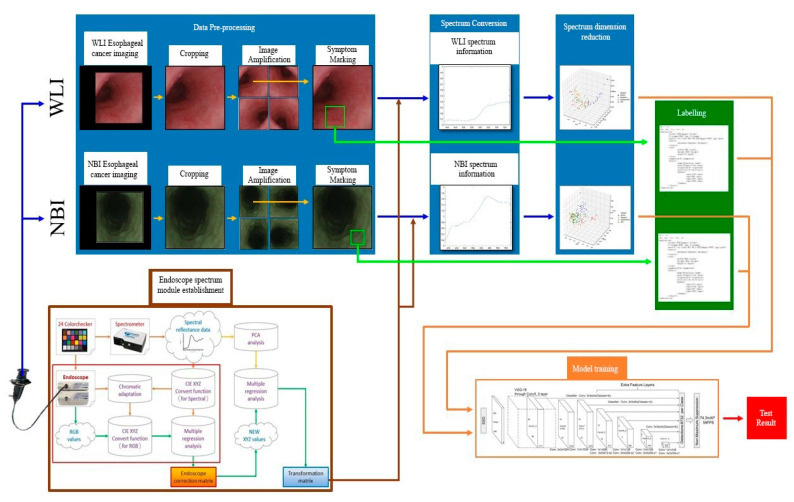
Flowchart of SSD esophageal cancer hyperspectral training.

**Figure 3 cancers-13-04593-f003:**
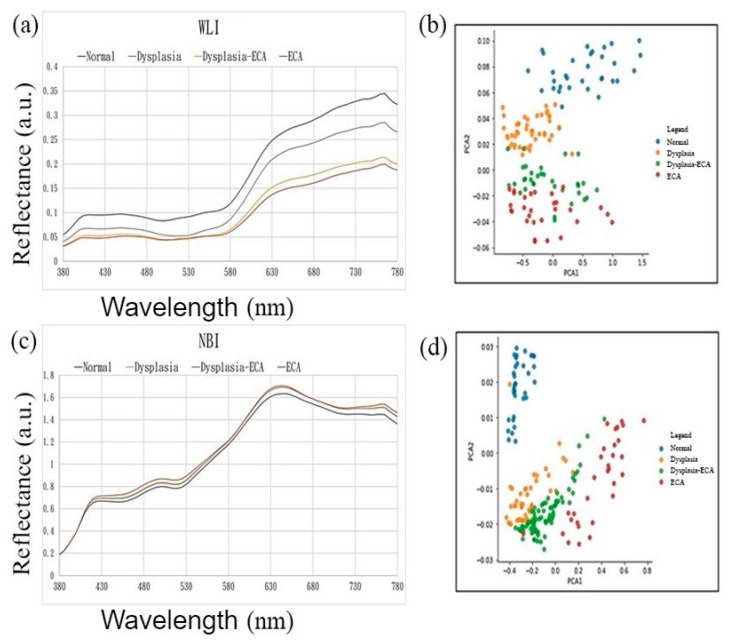
White-light and narrow-band endoscopic imaging of esophageal cancer spectrum and dimensionality reduction data distribution map. (**a**) White-light endoscopic images of esophageal cancer spectrum; (**b**) white-light endoscopic images of esophageal cancer spectrum dimensionality reduction data in the first two principal components of the data distribution map; (**c**) narrow-band endoscopic imaging of esophageal cancer spectrum; (**d**) narrow-band endoscopic imaging of esophageal cancer spectrum dimensionality reduction data in the first two principal components of the data distribution map.

**Figure 4 cancers-13-04593-f004:**
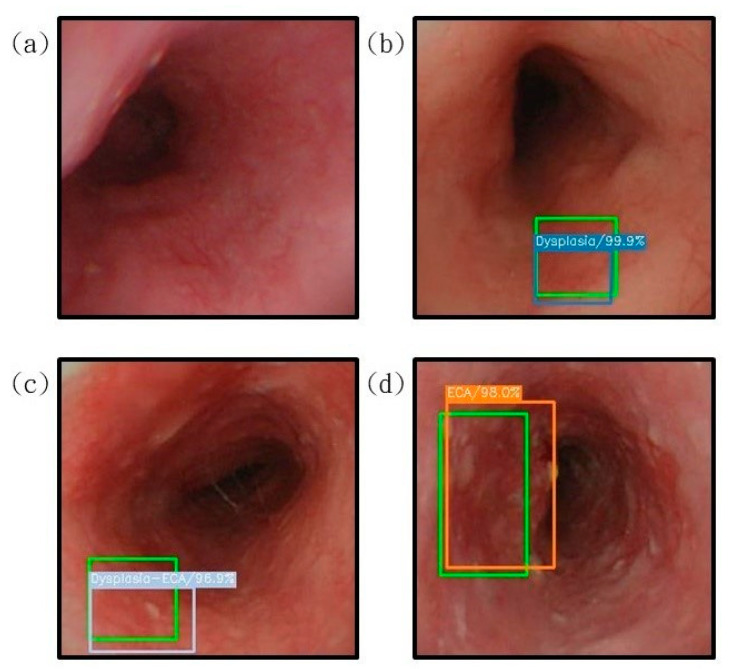
The results of the prediction of esophageal cancer on the white-light endoscopic imaging. The correct position is represented by green rectangle is shown for comparison with the predicted result. (**a**) The result of the esophagus test is normal. It is not displayed under the diagnosis of SSD. (**b**) represents the detection result of the area with low-grade dysplasia, under the diagnosis of SSD, the blue bounding box surrounds the lesion area; (**c**) shows the area with high-grade dysplasia. The test result shows that the gray bounding box surrounds the lesion area under the diagnosis of SSD; (**d**) shows the detection result of the invasive cancer area, and the orange bounding box surrounds the lesion area under the diagnosis of SSD.

**Figure 5 cancers-13-04593-f005:**
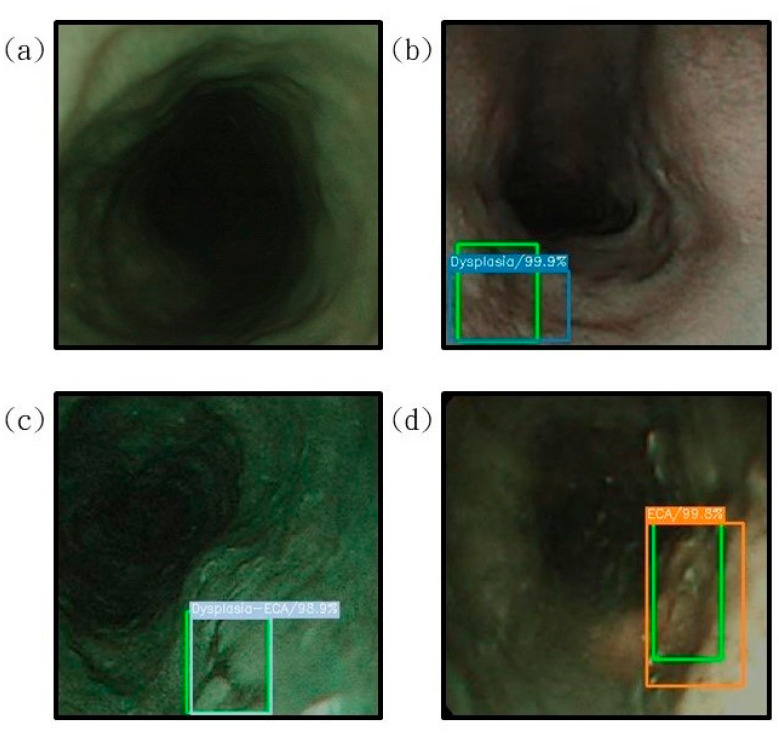
The results of the prediction of esophageal cancer are displayed on the narrow-band endoscopic imaging, where the correct position (**green rectangle**) is compared with the predicted result; (**a**) is the normal esophageal test result, which is not displayed under the diagnosis of SSD; (**b**) shows the detection result with low-grade dysplasia. Under the diagnosis of SSD, the blue bounding box surrounds the lesion area; (**c**) shows with the area between high-grade dysplasia and low-grade dysplasia. The test result shows that the gray bounding box surrounds the lesion area under the diagnosis of SSD; (**d**) shows the detection result of the invasive cancer area, and the orange bounding box surrounds the lesion area under the diagnosis of SSD.

**Table 1 cancers-13-04593-t001:** RGB image and spectrum data test results analysis.

WLI RGB	Sensitivity (%)	Precision (%)	F1-Score (%)	Accuracy (%)	Kappa
Normal	76	72	74	83	0.76
Low-grade dysplasia	61	85	71		
High-grade dysplasia	86	86	86		
Invasive cancer	93	85	89		
NBI RGB	Sensitivity (%)	Precision (%)	F1-score (%)	Accuracy (%)	Kappa
Normal	68	89	77	86	0.81
Low-grade dysplasia	83	88	85		
High-grade dysplasia	88	86	87		
Invasive cancer	98	84	90		
WLI spectrum	Sensitivity (%)	Precision (%)	F1-score (%)	Accuracy (%)	Kappa
Normal	92	85	88	88	0.84
Low-grade dysplasia	75	84	79		
High-grade dysplasia	92	88	90		
Invasive cancer	91	92	91		
NBI spectrum	Sensitivity (%)	Precision (%)	F1-score (%)	Accuracy	(%) Kappa
Normal	96	89	92	91	0.84
Low-grade dysplasia	89	92	91		
High-grade dysplasia	91	89	90		
Invasive cancer	91	94	93		

**Table 2 cancers-13-04593-t002:** Confusion matrix statistics table of RGB image and spectrum data test results, which compares the esophageal cancer spectrum data test and the test results using RGB esophageal cancer images.

Confusion Matrix		True Value				Total
WLI RGB		Normal	Dysplasia	High-Grade Dysplasia	Invasive Cancer	
Predicted value	Normal	13	3	0	2	112
	Low-grade dysplasia	1	11	1	0	
	High-grade dysplasia	3	1	30	1	
	Invasive cancer	0	3	4	39	
NBI RGB		Normal	Low-Grade Dysplasia	High-grade dysplasia	Invasive Cancer	
Predicted value	Normal	25	2	0	1	152
	Low-grade dysplasia	1	15	1	0	
	High-grade dysplasia	5	1	38	0	
	Invasive cancer	6	0	4	53	
WLI spectrum		Normal	Low-Grade Dysplasia	High-grade dysplasia	Invasive Cancer	
Predicted value	Normal	23	2	1	1	155
	Low-grade dysplasia	2	21	1	1	
	High-grade dysplasia	0	3	45	3	
	Invasive cancer	0	2	2	48	
NBI spectrum		Normal	Low-Grade Dysplasia	High-grade dysplasia	Invasive Cancer	
Predicted value	Normal	24	2	1	0	153
	Low-grade dysplasia	1	24	0	1	
	High-grade dysplasia	0	1	41	1	
	Invasive cancer	0	0	3	51	

## Data Availability

Data sharing not applicable.

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
