# Peer review of "Hyperspectral Imaging Combined with Artificial Intelligence in the Early Detection of Esophageal Cancer"

_cancers, 2021, doi:10.3390/cancers13184593_

Round 1

Reviewer 1 Report

The manuscript is well written but I have some query comments.

At the end of the Introduction you should write about the aim of the study.

Where the study was conducted?

Where the patients were examined?How many patients took part in the study?Please write in the discussion about the relevance of these results to clinical work. Moreover I think you should write if this method is used somewhere in everyday life in oncological diagnostics. If not, why not. Are there any methodological or hardware problems. Could you suggest an algorithm for incorporating this method into everyday clinical practice.

Reviewer 2 Report

Comments to the author

Overall impression

This is an interesting study investigating the accuracy of hyperspectral imaging combined with artificial intelligence deep learning methods in the early detection of esophageal cancer. The results of this study are very important and should be published, but some revision of the manuscript is necessary.

General remarks:

Use rounded figures throughout the paper including all tables and figures (e.g. 88% instead of 88.39%).

Change the term “dysplasia-ECA” throughout the paper including all tables and figures as it does not exist according to the WHO-classification. Please specify what you mean and correct (high-grade dysplasia? carcinoma in situ? microinvasive carcinoma? early cancer?). Hence, the 4 stages could include the following: normal, low-grade dysplasia, high-grade dysplasia, invasive cancer, or: normal, dysplasia, early cancer, invasive cancer.

Use the term “white-light (endoscopic) image” and “narrow-band (endoscopic) image” instead of “… endoscope”. Per definition, the endoscope (gastroscope) is an instrument, with produces the images and normally includes both WLI and NBI mode. However, in this study, the images are used for further processing (and not the endoscope). Use a uniform writing for “white-light” and “narrow-band” throughout the paper including all tables and figures.

Manuscript

Title: I would suggest a much shorter and concise title (e.g. hyperspectral imaging combined with artificial intelligence in the early detection of esophageal cancer)

Simple Summary: No changes are necessary except for rounded figures.

Abstract: Basically, no changes are necessary except for rounded figures. Although the author guidelines do not have strict formatting requirements, I personally think that a structured abstract (background, methods, results, conclusions) would be much clearer as one is forced to reduce the whole study to the most essential.

Introduction: This section is probably too long, and part of it rather belongs to the discussion. At the end of this section, a sentence with the objective of the study is missing. A logic structure would be the following: Why is early detection of esophageal cancer important? What is the problem of examiner-dependent interpretation of endoscopic images (experience, learning curve, intra- and interrater variability)? What is hyperspectral imaging and its advantages? What is artificial intelligence and its advantages (independent of examiner, helpful for beginners and experienced clinicians, lower intra- and interrater variability)? What was done so far in this field (keep the details for the discussion)? What is the objective of this study?

Methods: It is not clear to me, what was exactly done in this study. It seems that first the system was trained by 1232 images of esophageal cancer and its precursor lesions (620 WLI, 612 NBI), and then 308 images (155 WLI, 153 NBI) were evaluated with the trained system (accuracy of AI-HIS) ? If so, then clearly describe in this section, what you have done. In addition, the statistics of this study are missing in this section (they are shortly mentioned in the discussion but do not belong there). Please change the term “dysplasia-ECA” as it does not exist according to the WHO-classification (see also general remarks).

Results: The majority of this section belongs to the methods (except for the test results). Do not use the term “endoscope” when you mean “images” (see also general remarks). Table 2 summarizes all your results, thus the text should only mention the most essential and interpret your findings as you did in the discussion.

Discussion: The majority of this section belongs to the results as it is indeed a good summarization and interpretation of your results, whereas the section about kappa belongs to the methods. A logic structure would be the following: What is new in your study? What were your main findings? What have other research groups done so far (do not repeat the introduction, but give more details)? What were their findings? Are the latter corresponding to your findings or what could be the reasons for differences? What are possible problems or difficulties in your study? What could be done in the future to solve these problems (outlook for future studies)?

Conclusions: This section is probably too long, and only the last sentence is really necessary. However, the main part of this section could be used in the discussion.

References: The references in this paper are appropriate.

Abbreviations: The abbreviations used in this paper are appropriate. However, at first mention, they should be written out, and those occurring only once are not really necessary.

Paper Size

The paper size is adequate.

Tables and Figures

The tables and figures are adequate, but part of Figure 1 and its legend is missing, and the annotations in Tables 2-5 are partly too small to read and therefore should be corrected. In Table 2, among the Predicted Value for the NBI Spectrum, there should be “41” instead of “0” for correct estimated Dysplasia-ECA (as mentioned in the manuscript).
